# SAIPP-NET: A SAMPLING-ASSISTED INDOOR PATHLOSS PREDICTION METHOD FOR WIRELESS COMMUNICATION SYSTEMS

*Bin Feng*[*†]       *Meng Zheng*[‡]       *Wei Liang*[*]       *Lei Zhang*[*†]

[*] Shenyang Institute of Automation, Chinese Academy of Sciences, China
[†] University of Chinese Academy of Sciences, China
[‡] School of Computer Science and Engineering, Northeastern University, China

## ABSTRACT

Accurate prediction of pathloss radio maps is essential for the design and optimization of next-generation indoor wireless communication systems. Incorporating sparse pathloss measurements as auxiliary information has demonstrated significant potential in improving prediction accuracy. In this paper, we propose a novel sampling-assisted indoor pathloss prediction method (SAIPP-Net). First, we design a UNet-based neural network with variable-channel inputs to adapt to different levels of sampling availability. Second, we introduce a sampling-aware training strategy that employs tailored training schemes for low and high sampling rates, respectively. Finally, we develop a prioritized hybrid sampling strategy that jointly considers the transmitter distance and signal gradient to guide the selection of informative sampling locations. SAIPP-Net was evaluated in the context of *MLSP 2025 The Sampling-Assisted Pathloss Radio Map Prediction Data Competition*, achieving a weighted root mean squared error of 4.67 dB on the test set and securing 1st place in the competition.

***Index Terms***— Indoor pathloss prediction, sampling strategy, radio map, deep learning

## 1. INTRODUCTION

Accurate prediction of pathloss radio maps plays a pivotal role in the design, optimization, and management of wireless communication systems [1]. A pathloss radio map offers a spatial representation of large-scale signal attenuation in an environment, capturing the effects of obstacles, distance, and environmental characteristics on signal propagation. Traditionally, such maps are generated through extensive field measurements or computationally intensive ray-tracing simulations, both of which face the issue of scalability when being applied to large-scale or dynamic environments. To address these limitations, researchers have recently explored data-driven methods [2]. Levie et al. [3] demonstrate that well-designed and properly trained Deep Neural Networks (DNNs) can effectively estimate pathloss functions from readily available inputs such as urban maps and transmitter locations. For indoor environments, Bakirtzis et al. [4] propose a generalizable data-driven propagation model that incorporates wall permittivity and coarse-grained propagation priors—namely, empirical channel models defined by specific formulas (e.g., free-space pathloss model). These methods offer significant gains in efficiency and scalability, enabling real-time and low-cost radio map prediction.

While deep learning-based methods have shown promising results, they often neglect the valuable measurements available in practical deployments. In real-world scenarios, wireless communication systems are typically equipped with sparse sensors or mobile users that can report received signal strength at specific locations. These sampled measurements provide direct and reliable observations of the true pathloss in the environment, acting as strong anchors to guide and refine data-driven predictions. Integrating such sparse yet informative measurements into the prediction process is thus crucial for improving model accuracy and generalizability, particularly in heterogeneous and dynamic indoor environments.

Building upon this insight, *MLSP 2025 The Sampling-Assisted Pathloss Radio Map Prediction Data Competition* [5] was launched to promote deep learning methods for indoor pathloss radio map prediction with ground truth pathloss samples, with a particular focus on the role of sampling strategies. The competition includes two supervised tasks based on Indoor Radio Map Dataset [6]. Specifically, Task 1 is set to evaluate prediction performance for a fixed set of random samples at two sparsity levels (0.02% and 0.5%), while Task 2 allows participants to jointly optimize sampling locations and pathloss prediction subject to the same sampling constraints. The overarching goal of this competition is to assess how effectively different methods can exploit sparse measurements and choose sampling locations to improve prediction accuracy, while ensuring computational efficiency across diverse indoor environments.

In this paper, we propose a sampling-assisted indoor pathloss prediction method, termed as **SAIPP-Net**, to cope

This work was supported by the LRTP (XLYC2203148). Corresponding author: zhengmeng@cse.neu.edu.cn

with the two tasks in the competition. SAIPP-Net achieves a weighted Root Mean Square Error (RMSE) of 4.67 dB on the official test set and obtains the first overall ranking. The source code of SAIPP-Net is available at https://github.com/xsyl0011/SAIPP-Net. To summarize, the contributions of this paper are threefold:

1) **Network Design with Variable-Channel Inputs.** To accommodate varying levels of data availability, we design a UNet-based DNN model with variable-channel inputs. When no sampling data is available, the DNN takes as input a five-channel tensor encoding indoor geometry (reflectance and transmittance), transmitter distance, an augmented model channel, and the operating frequency band. Under high sampling rate conditions (i.e., 0.5%), an additional input channel representing the sampled pathloss values is appended to the five-channel tensor, forming a six-channel input. This novel input design allows SAIPP-Net to seamlessly adapt to different sampling scenarios, providing a unified foundation for the training strategies.

2) **Sampling-Aware Training Strategy.** SAIPP-Net employs a sampling-aware training strategy that adapts to different sampling rates. For a low sampling rate (0.02%), we first pre-train the model without using any sampled values, and then fine-tune it by using the sparse samples. In contrast, for a high sampling rate (0.5%), the sampled pathloss radio map is directly provided as an additional input channel, and the DNN is trained in an end-to-end learning manner. This dual-branch strategy ensures robust performance for both sparse and dense sampling settings.

3) **Prioritized Hybrid Sampling Strategy.** For Task 2, in which sampling locations can be actively selected, we introduce a Prioritized Hybrid Sampling Strategy (PHSS) that guides sampling based on both physical distance and signal variation. Specifically, PHSS assigns higher sampling probabilities to regions that are farther from the transmitter and to those with larger signal gradients (estimated from an initial radio map). Experimental results demonstrate that PHSS can select more informative samples and improve radio map prediction performance compared to random sampling.

## 2. PROBLEM FORMULATION

In this paper, we formulate indoor pathloss prediction as a deep learning-based image-to-image regression problem. In the considered indoor scenarios, each indoor environment is discretized into $H \times W$ grids. $H$ and $W$ vary between different indoor environments depending on their physical dimensions. Each indoor environment is represented as a multi-channel input image $\boldsymbol{X} \in \mathbb{R}^{C \times H \times W}$ that encodes key physical properties that can reflect electromagnetic propagation. The target output $\boldsymbol{Y} \in \mathbb{R}^{1 \times H \times W}$ is the ground-truth pathloss radio map generated by ray tracing simulations. Given training samples $[\boldsymbol{X}, \boldsymbol{Y}]$, the goal is to learn a mapping $f(\cdot)$ from

$\boldsymbol{X}$ to a pathloss radio map $\hat{\boldsymbol{Y}} \in \mathbb{R}^{1 \times H \times W}$, defined as

$$\hat{\boldsymbol{Y}} = f(\boldsymbol{X}|\Theta) \tag{1}$$

where $\Theta$ denotes the set of learnable weights. Each pixel in $\hat{\boldsymbol{Y}}$ represents the estimated signal attenuation (in dB) at the corresponding spatial location. The training process aims to find the optimal parameter set $\Theta^*$ that minimizes the discrepancy between $\hat{\boldsymbol{Y}}$ and the ground truth $\boldsymbol{Y}$.

## 3. PROPOSED SAIPP-NET METHOD

### 3.1. Input Feature Designing and Model Variants

The design of input features plays a critical role in the performance of deep learning models for radio map prediction. To improve prediction accuracy under different sampling settings, we consider two types of input configurations corresponding to two model variants, as illustrated in Fig. 1:

1) For the base model without samples, referred to as **IPP-Net**, we design a five-channel input configuration ($C = 5$). The first three channels correspond to the RGB image in the competition dataset [6]: the first two channels represent the reflection and transmission coefficients at each pixel, respectively; the third channel encodes the physical distance from the transmitter to each pixel location. The first two channels not only provide electromagnetic properties but also implicitly capture the indoor layout and material distribution. The third channel contains spatial information about the transmitter location. The fourth channel, referred to as the augmented model channel, is generated using a modified 3GPP InH model, providing a coarse but informative estimate of the radio propagation field. Notably, this channel improves upon the design in our previous work [7]. The fifth channel is the frequency channel that encodes the operating frequency band.

2) For the sampling-assisted model under high sampling rate circumstances (0.5%), referred to as **IPP-Net+**, we introduce a sixth channel representing the sampled pathloss map ($C = 6$). This additional input enables the model to incorporate partial ground-truth measurements, allowing it to leverage direct observations during supervised training.

### 3.2. Details of Augmented Model Channel Design

The wireless communication domain benefits from a wealth of theoretical models grounded in well-established electromagnetic principles. These models embed valuable domain knowledge that can be exploited to enhance the performance of radio map predictions. In particular, incorporating the outputs of empirical channel models as input feature maps provides strong inductive priors, which can improve the generalizability of deep learning models.

In this work, we adopt the 3GPP InH model as the baseline for our model channel design, as it is specifically developed for indoor wireless propagation scenarios. The standard

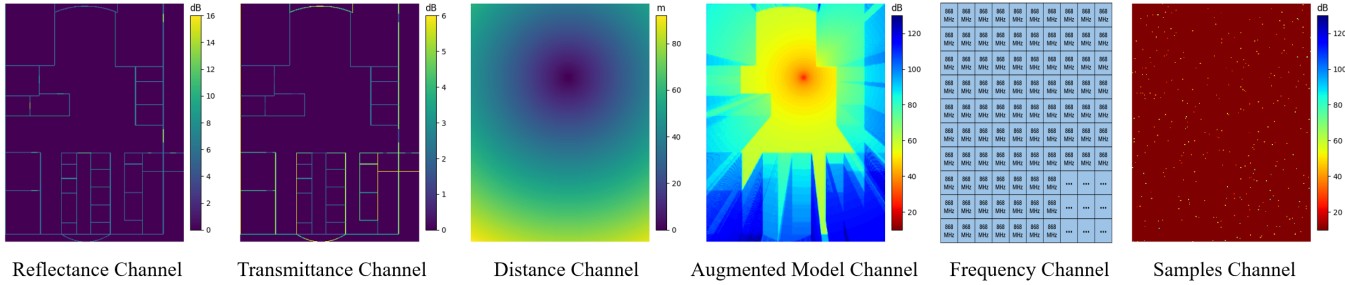

| Reflectance Channel | Transmittance Channel | Distance Channel | Augmented Model Channel | Frequency Channel | Samples Channel |

**Fig. 1**. Illustration of input configurations for IPP-Net (five-channel) and IPP-Net+ (six-channel) in SAIPP-Net

formulation of the 3GPP InH model in Line-of-Sight (LoS) and Non-Line-of-Sight (NLoS) scenarios is defined as [8]

$$
\begin{aligned}
PL_{\text{InH-LoS}} &= 32.4 + 17.3 \log_{10}(d_{3D}) + 20 \log_{10}(f_c), \\
PL'_{\text{InH-NLoS}} &= 17.3 + 38.3 \log_{10}(d_{3D}) + 24.9 \log_{10}(f_c), \\
PL_{\text{InH-NLoS}} &= \max\left(PL_{\text{InH-LoS}}, PL'_{\text{InH-NLoS}}\right),
\end{aligned}
\tag{2}
$$

where $d_{3D}$ is the three-dimensional Euclidean distance between the transmitter and a given location in the environment, and $f_c$ denotes the carrier frequency in GHz. However, the 3GPP InH model only distinguishes between LoS and NLoS scenarios in a coarse manner, and fails to capture the fine-grained propagation effects induced by material penetration, which are critical in indoor environments.

To address this limitation, we propose an augmented model channel that integrates a learnable material attenuation vector $\mathbf{\Delta}$ with material count map $\boldsymbol{M}_c$ into the standard 3GPP InH model. This enhancement enables the model to better fit the complex attenuation behavior caused by diverse wall and object materials. The details of the augmented model channel are as follows.

1) Material Count Map $\boldsymbol{M}_c$: We introduce a material count map $\boldsymbol{M}_c \in \mathbb{Z}^{N_m \times H \times W}$, where $N_m$ is the number of material types (or transmittance values). Each element $\boldsymbol{M}_c(n, h, w)$ records the number of times the straight line connecting the transmitter and the pixel location $(h, w)$ intersects material type $n$, based on a geometric computation over the transmittance channel. This is computed using Bresenham's line algorithm to trace the straight line between the transmitter and each pixel location, and count the materials intersected along the path. Based on the analysis of the competition dataset, the transmittance channel contains discrete values from the set $\{1, 2, 3, 4, 6, 10, 23\}$, corresponding to $N_m = 7$ material types.

2) Learnable Material Attenuation Vector $\mathbf{\Delta}$: We define a learnable material attenuation vector $\mathbf{\Delta} \in \mathbb{R}^{1 \times N_m}$, where each entry $\Delta_n$ represents the per-unit attenuation contributed by material type $n$. These parameters are optimized end-to-end during network training, allowing the model to adaptively learn how different materials affect signal attenuation.

3) Final Augmented Model Channel Representation: The final augmented model channel $\boldsymbol{M} \in \mathbb{R}^{1 \times H \times W}$ is con-

structed as

$$
\boldsymbol{M} = \boldsymbol{PL}_{\text{InH}} + \sum_{n=1}^{N_m} \Delta_n \boldsymbol{M}_c^{(n)},
\tag{3}
$$

where $\boldsymbol{PL}_{\text{InH}}$ is the pathloss radio map calculated by Equation (2), $\Delta_n$ denotes the attenuation coefficient for the $n$-th material type, and $\boldsymbol{M}_c^{(n)} \in \mathbb{Z}^{1 \times H \times W}$ is the material count map of the $n$-th material.

This design offers a more physically grounded and interpretable model channel input, which can guide the network to better learn the effects of transmission loss under heterogeneous indoor scenarios. This is particularly beneficial in indoor pathloss prediction tasks, where the refracted components of the electromagnetic field through walls and furniture play a more dominant role compared to the reflected components that often dominate in outdoor settings. Compared to our earlier version of the model channel proposed in prior work [7], which relied on a *NLoS level matrix* without distinguishing between materials, the current design offers a more fine-grained and learnable representation of material-induced attenuation.

### 3.3. Network Architecture

Convolutional neural networks constitute a foundational class of deep learning models for image processing tasks and are particularly well-suited to radio map prediction, which can be formulated as an image-to-image regression problem. Among them, UNet [9] has demonstrated strong performance across various dense prediction tasks, including medical image segmentation and image-to-image translation. Motivated by its prior success in pathloss prediction tasks [3, 4], we adopt a UNet-based architecture to approximate the mapping function $f(\cdot|\Theta)$ in Equation (1).

The network architecture (shown in Figure 2) is identical to the architecture in our previous work [7]. This choice is motivated by empirical evidence from past experiments showing that this architecture already achieves strong performance. Since the main focus of this competition is on incorporating and leveraging sampling data, we reuse the proven architecture and instead direct our innovation toward input feature design and sampling strategies.

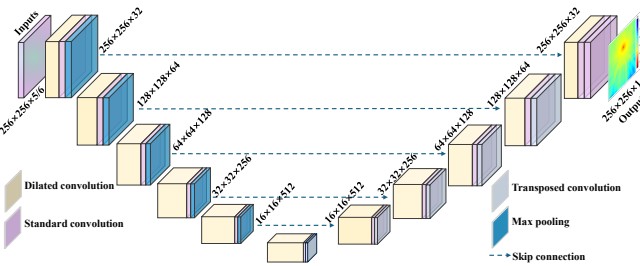

**Fig. 2**. Network architecture

### 3.4. Sampling-Aware Training Strategy

To effectively cope with the sub-tasks under different sampling rates in both Task 1 and Task 2, we design a **Sampling-Aware Training Strategy**. The core idea is to adapt the training procedure to the level of available sampling density—especially considering the large discrepancy between low and high sampling rate scenarios.

**1) Strategy for High Sampling Rate (0.5%)**

When the sampling rate is relatively high (0.5%), more abundant ground-truth pathloss values are available, enabling direct supervised learning. In this case, we adopt the IPP-Net+ variant, which augments the input tensor by including an additional channel to encode the sampled pathloss values. This additional input channel explicitly injects partial ground-truth into the model, allowing the model to directly utilize observed measurements during supervised training. This strategy not only helps the model learn to interpolate between observed samples but also enables better spatial generalization by preserving local signal patterns.

A two-stage curriculum training scheme is employed for IPP-Net+:

**Stage 1**: Train the model on multiple indoor environments at a single frequency (868 MHz).

**Stage 2**: Further train the model on multi-frequency data (868 MHz, 1.8 GHz, and 3.5 GHz).

Although the test set includes only 868 MHz data, leveraging multiple frequencies enhances model generalization by introducing greater data diversity and exploiting shared propagation characteristics across frequencies.

**2) Strategy for Low Sampling Rate (0.02%)**

In contrast, when the sampling rate drops to a very low level (0.02%), only extremely sparse ground-truth pathloss values are available, which makes the direct training of a DNN model from scratch highly prone to overfitting. A straightforward and robust approach is using samples to fine-tune a pre-trained model. As introduced earlier, IPP-Net takes a five-channel input that does not include the samples channel. We first pre-train the base IPP-Net using the aforementioned two-stage curriculum learning strategy and then fine-tune it using the sparse 0.02% samples. We adopt a full fine-tuning strategy and introduce a masked loss function

$\mathcal{L}_{\text{mask}}$ to supervise only the sampled locations:

$$
\mathcal{L}_{\text{mask}} = \sqrt{\frac{\sum_{h=1}^{H} \sum_{w=1}^{W} \mathcal{M}_{h,w} \cdot \left(\hat{\boldsymbol{Y}}_{h,w} - \boldsymbol{Y}_{h,w}\right)^2}{\sum_{h=1}^{H} \sum_{w=1}^{W} \mathcal{M}_{h,w}}}, \quad (4)
$$

where $\mathcal{M} \in \{0, 1\}^{H \times W}$ is a binary sampling mask. $\mathcal{M}_{h,w} = 1$ indicates that pixel $(h, w)$ has a sampled value, and $\mathcal{M}_{h,w} = 0$ otherwise.

### 3.5. Prioritized Hybrid Sampling Strategy

To better exploit sparse sampling resources, we propose PHSS (Algorithm 1) which combines both physical-domain knowledge and data-driven insights to optimize sampling locations for radio map prediction. By targeting regions that are either physically significant or exhibit high signal variation, PHSS enables a more efficient allocation of limited sampling budgets.

**Distance-Based Sampling (30% budget):** In radio propagation, signal strength typically exhibits greater variation at locations farther from the transmitter. To capture the global propagation structure, this stage assigns sampling probabilities proportional to the logarithm of the transmitter distance, encouraging spatially distributed sampling.

**Gradient-Based Sampling (70% budget):** Based on an initial radio map estimated by IPP-Net, this stage computes gradient magnitudes to identify areas with sharp signal transitions. Sampling probabilities are assigned proportionally to the gradient values, prioritizing regions with high spatial variation, such as boundaries between propagation zones.

## 4. EXPERIMENTS AND RESULTS

### 4.1. Experimental Setup

All experiments were implemented using the PyTorch framework on an NVIDIA RTX 3080 Ti GPU. The competition dataset [6] was split into training and validation sets with a ratio of 9:1. All input data are resized to $256 \times 256$ to enable mini-batch training. Standard data augmentation techniques, including random rotations and horizontal/vertical flips, are applied to the training set to improve generalizability. Table 1 summarizes the key hyperparameters used throughout the experiments.

### 4.2. Final Evaluation under different settings

The final evaluation was conducted by the competition organizers on an unseen test set comprising five indoor scenarios, collectively containing 200 radio map instances. SAIPP-Net was evaluated under multiple fine-tuning strategies. The numerical results and run-times are summarized in Table 2. Results show that fine-tuning on the entire test dataset leads to lower RMSE but may exploit prior information unavailable in real deployment. In realistic deployment scenarios, the target environment is typically assumed to be unseen during

**Algorithm 1:** Prioritized Hybrid Sampling Strategy

---

**Input:** Distance channel $D_{\text{tx}}$; target sampling rate $\rho$; radio map size $H \times W$

**Output:** Sampling mask $\mathcal{M} \in \{0,1\}^{H \times W}$

---

**1** **Distance-Based Sampling (30% budget)** ;

**2** Compute normalized log-distance weights:

**3** $\quad W_d \leftarrow \dfrac{\log(1 + D_{\text{tx}} + \epsilon)}{\max(\log(1 + D_{\text{tx}} + \epsilon))}$ ;

**4** Compute budget: $n_1 \leftarrow \lfloor \rho \cdot H \cdot W \cdot 0.3 \rfloor$ ;

**5** Sample $n_1$ locations with probability $\propto W_d$ to obtain index set $\mathcal{I}_1$ ;

**6** **Gradient-Based Sampling (70% budget)** ;

**7** Estimate radio map $\hat{Y}$ using IPP-Net ;

**8** Compute gradient magnitude:

**9** $\quad G \leftarrow \sqrt{(\nabla_x \hat{Y})^2 + (\nabla_y \hat{Y})^2}$ ;

**10** Set $G[i,j] \leftarrow 0, \forall(i,j) \in \mathcal{I}_1$ ;

**11** Compute remaining budget: $n_2 \leftarrow \lceil \rho \cdot H \cdot W \rceil - n_1$ ;

**12** Sample $n_2$ locations with probability $\propto G$ to obtain index set $\mathcal{I}_2$ ;

**13** **Construct Final Sampling Mask** ;

**14** Initialize $\mathcal{M} \leftarrow \mathbf{0}^{H \times W}$ ;

**15** Set $\mathcal{M}[i,j] \leftarrow 1, \forall(i,j) \in \mathcal{I}_1 \cup \mathcal{I}_2$ ;

**16** **return** $\mathcal{M}$

---

**Table 1**. Experiment settings

| Hyperparameter | Value |
|---|---|
| Learning rate | $10^{-3} \sim 3.125 \times 10^{-5}$; $10^{-4}$ (fine-tuning) |
| ReduceLROnPlateau | factor = 0.5, patience = 5 |
| Batch size | 8 |
| Optimizer | AdamW, weight decay = $10^{-2}$ |
| Max epochs | 120; 2 (fine-tuning) |
| Loss function | RMSE |

training, including any prior samples from that environment. Fine-tuning on the full test set may inadvertently introduce information leakage across transmitter locations within the same environment, thus potentially inflating performance and undermining the evaluation of model generalizability. Therefore, the official scores are based on fine-tuning individually on each test radio map instance for 2 epochs, which is more aligned with the competition's motivation and runtime requirements. Under this setting, SAIPP-Net achieves an RMSE of 5.99 and 6.08 dB on Task 1 and Task 2, respectively. The final score (4.67 dB) is calculated as a weighted average: $0.3 \times (5.99 + 3.32) + 0.2 \times (6.08 + 3.28) = 4.67$.

In addition to the official test set, the organizers also provided a test subset on the *Kaggle* website. This subset contains 50 radio maps from two unseen scenarios and serves as a publicly accessible benchmark for model evaluation. Figure 3 presents several radio maps predicted by SAIPP-Net from the Kaggle test subset, illustrating SAIPP-Net's ability to capture fine-grained spatial variations in pathloss distribution. Despite the extremely low sampling rate of 0.02%, SAIPP-Net can still recover the general propagation patterns, while higher sampling density (0.5%) significantly improves the prediction of local details and sharp transitions. This demonstrates the model's robustness under severe sparsity and its effectiveness in leveraging minimal ground-truth values to reconstruct high-fidelity pathloss maps.

### 4.3. Ablation Studies

We conducted a series of ablation studies to evaluate the impact of different training strategies and model configurations. All comparison results reported below are based on the Kaggle test subset on Task 1, to ensure fair and consistent comparisons across methods.

The comparison results on training strategies are summarized in Table 3. For the extremely sparse setting (0.02%), fine-tuning leads to significantly better performance than using samples as input, which even underperforms the base model without any samples. In contrast, under the high sampling rate of 0.5%, incorporating samples as input achieves the best performance, outperforming fine-tuning. This discrepancy likely stems from the fact that sparse samples at 0.02% are highly random and may not reflect meaningful propagation patterns, thus acting like noise when used as input. Fine-tuning, on the other hand, utilizes these values more cautiously through supervised loss, leading to better adaptation without hurting generalization.

The comparison results between different models under the sampling rate of 0.5% are summarized in Table 4. We observe that the augmented model channel $M$ contributes to performance improvement, and also outperforms the model channel in our previous work [7]. However, relying solely on the extracted features $M$—while discarding the reflectance, transmittance, and distance channels—proves inadequate and leads to performance degradation. Furthermore, our experiments reveal that directly using the radio map generated by $M$ as the final prediction (computed via Equation (3) with the learned material attenuation vector $\mathbf{\Delta}$=[-0.3214, -0.1678, 2.2897, 0.9304, 5.4183, 2.7411, 4.1387]) results in significant performance gaps. This suggests that while $M$ does enhance performance, Equation (3) alone cannot adequately capture the patterns of electromagnetic wave propagation.

### 5. CONCLUSION

In this paper, we proposed SAIPP-Net, a sampling-assisted indoor pathloss prediction method for wireless communication systems. SAIPP-Net integrated a UNet-based backbone with variable-channel inputs, a sampling-aware training strategy, and a prioritized hybrid sampling strategy. Extensive experiments demonstrated that SAIPP-Net can effectively leverage both sparse and dense sampling data to enhance prediction accuracy across different indoor scenarios. SAIPP-Net achieved a weighted RMSE of 4.67 dB and secured 1st place

**Table 2**. Performance and run-time of SAIPP-Net under different settings on official test set

| Sampling Rate | Fine-tuning Strategy | Task 1 RMSE (dB) | Task 2 RMSE (dB) | Run-time (ms) |
|---|---|---|---|---|
| 0.02% | 1 epoch (entire test dataset) | 5.86 | 5.70 | 60 |
| 0.02% | 1 epoch (per radio map) | 6.35 | 6.39 | 60 |
| 0.02% | 2 epochs (entire test dataset) | 5.65 | 5.47 | 106 |
| 0.02% | 2 epochs (per radio map) | 5.99 | 6.08 | 106 |
| 0.5% | None | 3.32 | 3.28 | 18 |
| No samples | None | 6.90 | 6.90 | 18 |

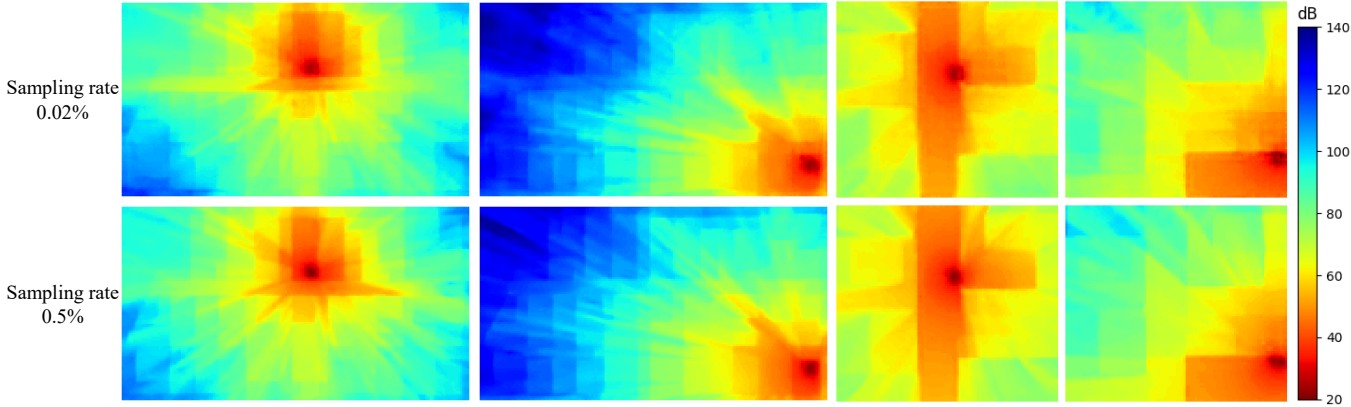

Sampling rate 0.02%

Sampling rate 0.5%

**Fig. 3**. Visualization of predicted pathloss radio maps on the Kaggle test subset (Task 1)

**Table 3**. Comparison on training strategies

| Sampling Rate | Training Strategy | RMSE |
|---|---|---|
| No samples | N/A | 5.9657 |
| 0.02% | Samples as input | 6.4394 |
| | Samples for fine-tuning | **5.3663** |
| 0.5% | Samples as input | **3.7224** |
| | Samples for fine-tuning | 4.5964 |

**Table 4**. Comparison between different models

| Model Configuration | RMSE |
|---|---|
| SAIPP-Net | **3.7224** |
| SAIPP-Net without $M$ | 4.1975 |
| IPP-Net [7] with samples channel | 4.1057 |
| Only $M$ and samples channel | 4.6207 |
| $M$ as final prediction | 9.9447 |

in *MLSP 2025 The Sampling-Assisted Pathloss Radio Map Prediction Data Competition*.

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
