# OpenReview forum: "SAIPP-Net: A Sampling-Assisted Indoor Pathloss Prediction Method for Wireless Communication Systems"
_IEEE.org/MLSP/2025_SA_Radio_Map_Prediction_Challenge — SA Radio Map Prediction Challenge at MLSP 2025 Oral_

### Official Review · Reviewer_yyme · 2025-06-05
**Review for: ``SAIPP-Net: A Sampling-Assisted Indoor Pathloss Prediction Method for Wireless Communication Systems''**

**Rating:** 9
**Confidence:** 5

**Review:**

This paper proposes SAIPP-Net, a sampling-assisted indoor pathloss prediction method designed for radio map generation using deep learning. The authors introduce three main contributions: a UNet-based model with variable-channel inputs, a sampling-aware training strategy, and a hybrid sampling approach combining distance-based and gradient-based sampling. The method is evaluated on the MLSP 2025 Sampling-Assisted Pathloss Radio Map Prediction Competition and secures 1st place.

The paper is well-organized and technically sound. The model design demonstrates a clear understanding of wireless propagation characteristics and effectively integrates both domain knowledge and data-driven techniques. The sampling strategies are carefully tailored to the specific constraints of each task and sampling rate. However, a few issues in clarity should be addressed before publication.


1. The introduction of terms like "IPP-Net", “IPP-Net+'', ”IPP-Net++'' and "SAIPP-Net" creates unnecessary naming complexity, given that all variants share the same network architecture. A more consistent and transparent naming convention would improve readability and avoid misleading the reader into thinking these are structurally distinct models.

2. Fig. 3 does not include a ground truth comparison, which is critical for interpreting pathloss prediction quality.

---

### Official Review · Reviewer_FHyw · 2025-06-05

**Rating:** 9
**Confidence:** 4

**Review:**

The paper is clearly written and engaging. The results are meaningful and well-presented. The only improvement that I can think of is in the introduction: maybe the authors could consider adding some text regarding the problem at hand in general. Along with some SOTA references. So that even the uninformed reader gets the catch early on.

---

### Official Review · Reviewer_ggcQ · 2025-06-07
**SAIPP-Net++: Intricately Designed Prior Feature coupled with Insightful Training/Fine-Tuning Strategy**

**Rating:** 9
**Confidence:** 4

**Review:**

The paper writes clearly about the feature design, hybrid sampling scheme, and the tailored training/fine-tuning strategies from the 1st-ranked method in the competition. The motivation behind the methodology is well-explained and easy to follow, supported by experimental results. Following are some questions about the technical details that might help the readers to better understand the method:

1. In the section of introduction, does "coarse-grained propagation priors" mean some models as the 3GPP InH model mentioned later in the paper?
2. Also in the introduction section, "uniformly random samples" might be misleading. I understand that "uniform" means the sampling probability is uniform across all candidates, however, it can also imply that the sampled points are uniformly located on the plane with equal distance etc.,
3. For the material count map Mc, it seems like the Bresenham's line algorithm is run for N_m times, which seems computationally demanding. Did you compare the runtime of generating the feature with the inference time? Moreover, Here, different material is identified by the transmittance values only, if transmittance-reflectance tuple is used to identify materials, N_m might be even greater.
4. For the gradient-based sampling, is it ever risky that the algorithm will output sampling points concentrated in certain area where the scale of the gradients is large?